# Impact of Two Commercial *S. cerevisiae* Strains on the Aroma Profiles of Different Regional Musts

**Francesca Patrignani** [1,2], **Gabriella Siesto** [3,*], **Davide Gottardi** [1,2,*], **Ileana Vigentini** [4], **Annita Toffanin** [5], **Vasileios Englezos** [6], **Giuseppe Blaiotta** [7], **Francesco Grieco** [8], **Rosalba Lanciotti** [1,2], **Barbara Speranza** [9], **Antonio Bevilacqua** [9] and **Patrizia Romano** [10]

[1] Department of Agricultural and Food Sciences, University of Bologna, 47521 Cesena, Italy
[2] Interdepartmental Center for Industrial Agri-food Research, University of Bologna, 47521 Cesena, Italy
[3] School of Agricultural, Forestry, Food and Environmental Sciences, University of Basilicata, 85100 Potenza, Italy
[4] Department of Food, Environmental and Nutritional Sciences, University of Milan, 20133 Milan, Italy
[5] Department of Agriculture, Food and Environment, University of Pisa, 56124 Pisa, Italy
[6] Department of Agricultural, Forest, and Food Science, University of Turin, 10095 Turin, Italy
[7] Department of Agricultural Sciences, Division of Grape and Wine Sciences, University of Naples Federico II, 83100 Avellino, Italy
[8] Institute of Sciences of Food Production, National Research Council of Italy, 73100 Lecce, Italy
[9] Department of Agriculture, Food, Natural Resources and Engineering, University of Foggia, 71122 Foggia, Italy
[10] Faculty of Economy, Universitas Mercatorum, 00186 Rome, Italy
* Correspondence: gasiesto@gmail.com (G.S.); davide.gottardi2@unibo.it (D.G.); Tel.: +39-366-7276643 (G.S.); +39-0547-338145 (D.G.)

**Abstract:** The present research is aimed at investigating the potential of two commercial *Saccharomyces cerevisiae* strains (EC1118 and AWRI796) to generate wine-specific volatile molecule fingerprinting in relation to the initial must applied. To eliminate the effects of all the process variables and obtain more reliable results, comparative fermentations on interlaboratory scale of five different regional red grape musts were carried out by five different research units (RUs). For this purpose, the two *S. cerevisiae* strains were inoculated separately at the same level and under the same operating conditions. The wines were analyzed by means of SPME-GC/MS. Quali-quantitative multivariate approaches (two-way joining, MANOVA and PCA) were used to explain the contribution of strain, must, and their interaction to the final wine volatile fingerprinting. Our results showed that the five wines analyzed for volatile compounds, although characterized by a specific aromatic profile, were mainly affected by the grape used, in interaction with the inoculated *Saccharomyces* strain. In particular, the AWRI796 strain generally exerted a greater influence on the aromatic component resulting in a higher level of alcohols and esters. This study highlighted that the variable strain could have a different weight, with some musts experiencing a different trend depending on the strain (i.e., Negroamaro or Magliocco musts).

**Keywords:** volatile molecule profile; *Saccharomyces cerevisiae* EC1118; *S. cerevisiae* AWRI796; regional grape musts; interlaboratory scale fermentation

## 1. Introduction

The chemical and aromatic composition of wine can be described as the result of many factors, including grape variety, geographical origin, viticultural condition of grape cultivation, natural occurring microbial ecology of the grape, fermentation process, and winemaking practices [1,2]. It is widely recognized that the metabolic characteristics of a particular yeast strain may lead to the formation of molecules that may affect the sensorial characteristics of wine [3]. Therefore, to ensure reliable and reproducible fermentations, most vinifications are carried out with the inoculation of selected strains of commercial

or indigenous yeasts [4], using the huge patrimony of naturally present microorganisms as a source of biodiversity [5]. Over the years, *Saccharomyces cerevisiae* wine strains have been isolated from natural fermentations, demonstrating the existence of a strong polymorphism within this species [6,7]. Each strain has its own metabolic characteristics, different from those of the other strains, exerting a profound influence on the aromatic balance of wine [8–13]. Several studies have demonstrated that different strains of *S. cerevisiae*, employed as starters, produce strain-specific metabolites [14,15] and can impact the chemical and aromatic profiles of wine, resulting in greater complexity and aromatic diversity [1]. Differences in the diversity of wines may also depend on the cultivar applied, the winery, and the fermentation conditions [16]. However, the correct use of microbial biodiversity remains one of the most important tools to increase aroma compounds in wine [17]. The differences observed appear to be more quantitative than qualitative, as suggested by Torrens et al. [18] who showed how yeast strain quantitatively impacted the chemical and volatile composition of Cava base wines more than must variety and harvest year.

According to the literature, wine aroma is a very complex concept since it is the result of the interaction of hundreds of different compounds with concentrations ranging between ppb and ppm. Their balance and interaction define wine aromatic quality [17,19]. According to their origins, wine aromas are traditionally divided into three main categories: varietal compounds deriving from grapes (primary aromas), aromas resulting from fermentation (secondary aromas), and compounds deriving from aging in barrels and bottles (tertiary aromas). Alcoholic fermentation, in addition to ethanol and carbon dioxide, produces hundreds of compounds which have a strong impact on the overall wine aroma [20,21] and range from µg to g/L [22,23]. This process increases the chemical and flavor complexity of wine, either by assisting in the extraction of compounds from their precursors present in grape must or modifying some grape derived compounds. Consequently, the selection criteria for starter cultures have been focused on the production of different metabolites which form the "wine fermentation bouquet", mainly secondary compounds such as higher alcohols and esters [24,25]. However, wine contains many other types of compounds, such as carbonyls, acids, terpenes, norisoprenoids, and sulfur compounds [26,27]. Volatile organic compounds (VOCs), with different polarities and volatilities and produced in different concentrations, have an important aromatic value and play a central role in defining wine sensorial identity [22,23]. Each category of aromatic compounds varies considerably among different types of wines with different predominant aromas which confer specific typicity on each wine [26–28]. *S. cerevisiae* is primarily responsible for the formation of higher alcohols and esters [29,30]. In regards to higher alcohols, they are generally considered to be aromatic molecules with the strongest effect on the global wine aroma and can have positive or negative sensory impacts, depending on their concentration. Values greater than or equal to 400 mg/L may result in a pungent aroma, while higher alcohols below 300 mg/L often impart desirable fruity characteristics [31]. Among them, isoamyl alcohol is the most represented (over 50%) and the main responsible compound for the fragrant component. Strains of *S. cerevisiae* can produce high amounts of isoamyl alcohol in a strain-specific fashion [14]. After higher alcohols, esters are the second most important component of volatile aromas in wine. They are naturally produced by yeasts during alcoholic fermentation through esterification of alcohol and low pH acids, and they directly contribute to the floral and fruity characteristics associated with the final product [32]. Their amount varies significantly and depends on several factors, such as grape cultivar, amino acids present in the must, and particularly the yeast strain applied [3]. In mixed fermentation, higher alcohols and esters may differ depending on the dominance of two or more strains [15,33]. Carbonyl compounds, such as aldehydes and ketones, are also produced during fermentation and play an important role in the final aroma profile of wine. The most common and represented aldehyde (90%) is acetaldehyde. Additionally, in this case, aldehydes and ketones production are strain-specific processes that require the selection of suitable strains according to the type of wine produced [34,35]. Finally, yeasts produce several organic acids that play a significant role in wine sensory perception

and directly impact the overall organoleptic characteristics of the product [36]. In the case of *S. cerevisiae*, the organic acid concentration and composition also changes significantly during alcoholic fermentation depending on the strain applied, with some of them being higher producers of organic acids in white and/or red wine fermentations, while others are generally lower organic acid producers [37].

Our experimental design, promoted by the Italian Group of Microbiology of Vine and Wine (GMVV), was planned to eliminate the effect of all the variables of the fermentation process, focusing the study on the interaction between *S. cerevisiae* strain and grape variety, with the aim of coupling a certain aromatic result with a typical strain behavior. The differential impact of two commercial *S. cerevisiae* wine strains, EC1118 and AWRI796, commonly known among oenologists and yeast producers, was assessed on the aroma profile of five different regional grape musts. To limit the variables in the fermentation process and obtain more reliable results, interlaboratory scale comparative fermentations of five different regional red grape musts were carried out by five different research units (Rus) under identical conditions and following the same protocol during the entire process [38].

## 2. Materials and Methods

### 2.1. Strains

The experiments were performed with two *S. cerevisiae* commercial strains, Lalvin EC1118 (Lallemand Inc., Montreal, QC, Canada) and AWRI796 (MAURIVIN, Tebaldi, Varese, Italy). Throughout the experiments and data, yeasts were labeled as EC and AW, respectively. Yeasts were purchased in the form of active dry yeast (ADY). To eliminate the variable linked to the strain inoculation, the same batch of each commercial yeast was used for all the fermentations carried out by the five Rus.

### 2.2. Inoculum Preparation

Inoculum was prepared according to the resolution OIV OENO 329/2009 [39]. Dry strains (1 g) were suspended under aseptic conditions in 100 mL of water at 36–40 °C, also containing 5% sucrose. Yeast suspension was slowly homogenized for 5 min two times, with a rest time between the two steps of 20 min. After preparation, yeast cell density was assessed through Thoma hemocytometer to gain an inoculum for fermentation at $2 \times 10^6$ cell/mL.

### 2.3. Fermentation Trials

Fermentations were carried out by 5 different Rus on Sg (Sangiovese), Ma (Magliocco), Ba (Barbera), Cs (Cabernet-Sauvignon), and Ne (Negroamaro) in 500-mL flasks, containing 350 mL of each natural red grape must. In order to avoid the development of the spontaneous microbiota of the grape must and to allow the inoculated strain to dominate the fermentation process, sulfur dioxide was added as potassium metabisulfite (stock solution, 10 g/L) to gain a concentration of 20 mg/L of $SO_2$, together with a massive inoculum of each *S. cerevisiae* strain. After strain inoculation, the presence of viable cells was verified by plate counting on Wallerstein Laboratory Nutrient Agar medium (WL, Oxoid, Hampshire, UK [40]).

YAN (Yeast Assimilable Nitrogen) was adjusted to 7.5 mg/L of YAN per 10 g/L of initial sugars, using a stock solution of 10 g/100 mL of "Supervit" (Enartis, Novara, Italy) corresponding to 20 mg/mL of YAN. YAN was quantified using Megazyme kit (Wicklow, Ireland), as reported by Boudreau et al. [41]; residual sugars were assessed by a Fourier Transfer Infrared WineScan instrument (FOSS, Hillerød, Denmark).

After yeast inoculation, flasks were closed through a Muller valve containing sulfuric acid up to the height of the internal glass tube, and incubated at 25 ± 2 °C in static conditions; fermentation lasted until sugar depletion.

For each must, fermentations were carried out in triplicate on three different batches for each strain (6 fermentations per each RU).

### 2.4. Volatile Aroma Profile

VOCs were evaluated through solid phase micro-extraction—gas chromatography—mass spectrometry (SPME-GC/MS), through a GC/MS Shimadzu QP2010 (Shimadzu, Kyoto, Japan) equipped with an auto sampler AOC-5000 plus and an SPME fiber coated with carboxen/polydimethylsiloxane (CAR/PDMS) phase (65 μm, SUPELCO, Bellefonte, PA, USA).

Before analysis, 5 mL of sample were added with 4-methyl 2-penthanol (internal standard, 100 mg/L) and 0.5 g of NaCl and sealed in a sterile vial.

The conditions for analysis (sample hating, adsorbing, and desorbing of VOCs, as well as flowrate and temperatures) were set as detailed in Romano et al. [38]. VOCs were identified through the NIST 8.0 (National Institute of Standards and Technology, Gaithersburg, MD, United States) library, while the assessment of their amount (equivalent mg/L) was obtained thanks to the internal standard.

### 2.5. Statistical Analysis

The experiments were performed over three independent samples; before the statistical analysis, homoscedasticity was checked. The first analysis was two-way analysis of variance (ANOVA) of both the total amounts of classes of compounds (esters, alcohols, acids, aldehydes, and ketones), and the amounts of some selected compounds; the categorical predictors were the strain (EC or AW) and the type of must (Sg, Ma, Ba, Cs, or Ne). *p*-level was set to 0.05, as dependent variables.

The outputs of two-way ANOVA were:

(a) The table of standardized effects, which reports on the statistical weight (Fisher test and *p*-value) of the kind of strain, or must, as well as on their interaction.

(b) The decomposition of the statistical hypothesis, which is a mathematical function showing the correlation of each factor (must, strain, must × strain) with the dependent variable.

Then, data (total amounts of VOCs, and actual amounts of some selected compounds) were analyzed through two way-joining, based on single linkage approach and percent disagreement; in order for the analysis to focus on data reliability, the independent batches were used as different samples. Finally, data were analyzed through Principal Component Analysis (PCA, Euclidean linkage approach) both as total amounts of classes and as selected compounds; the average values of the three replicates were used. Statistical testing was performed through the software Statistica for Windows (Statsoft, Tulsa, OK, USA).

### 3. Results

Regional grape musts (Sangiovese, Magliocco, Barbera, Cabernet, and Negroamaro) were used by five RUs as fermentation substrates for the two *S. cerevisiae* strains (EC1118 and AWRI796). To avoid the development of indigenous yeasts naturally occurring on the grapes, and to ensure that the fermentations were carried out by the inoculated strains, yeast isolation was performed after strain inoculation. The isolated colonies showed the morphology of *Saccharomyces*, at the concentration deliberately inoculated.

Wines obtained from each RU were analyzed at the end of fermentation by means of SPME-GC/MS. This technique allowed the identification of about 150 molecules belonging to different chemical classes, including alcohols, esters, organic acids, aldehydes, ketones, and other minor compounds such as terpenes and sulfur compounds.

According to Figure 1, reporting the two-way joining run for esters and alcohol (Figure 1A) and acid, ketones and aldehydes (Figure 1B) produced in the different samples (3 independent repetitions for each considered must and strain), the amount of molecules detected was different according to the *S. cerevisiae* strain used and the must employed. Alcohols and esters were produced in a range between 20 and 200 equivalent ppm while acids, aldehydes, and ketones were present in a range between 2 and 8 ppm. The most abundant alcohols were detected in wines obtained from Magliocco (Ma), independently from the strain applied, and in wines produced from must Negroamaro (Ne), fermented

with *S. cerevisiae* AW. On the contrary, samples from Negroamaro fermented with *S. cerevisiae* EC were characterized by a lower number of alcohols. Regarding esters, the highest production was found in samples from must Magliocco fermented with AW and must Negroamaro fermented with both the strains. Regarding acids, the major production was observed in samples from must Negroamaro and must Sangiovese (Sg) produced with *S. cerevisiae* AW and EC, respectively. Aldehydes and ketones were detected in lower amounts with respect to acids, especially in Magliocco and Barbera (Ba) fermented with *S. cerevisiae* EC and Barbera and Negroamaro fermented with strain AW.

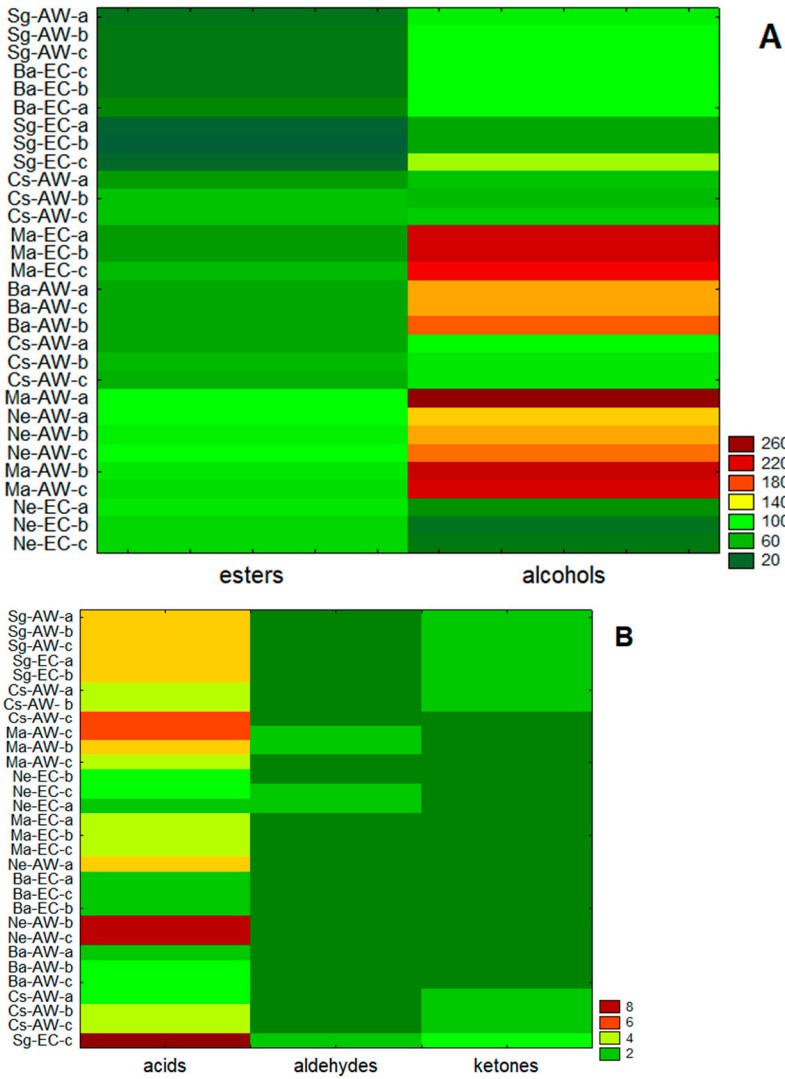

**Figure 1.** Two way-joining run on the total amounts (ppm) of esters, alcohols (**A**), and acids, aldehydes and ketones (**B**). Sg, Sangiovese; Ma, Magliocco; Ba, Barbera; Cs, Cabernet-Sauvignon; Ne, Negroamaro. EC, strain EC1118; AW, strain AWRI796.

Furthermore, Table 1 shows that the production of volatile compounds is significantly affected by the initial composition of the must (for all the compounds produced, but especially for esters, alcohols and ketones), by the variable strain (for esters and alcohols) and by the interaction must × strain. Since alcohols, esters, and acids represent overall the volatile compounds affecting the sensorial features of wine, the significance of the variables must, strain, and strain × must was investigated deeply.

**Table 1.** Significance (*p*-level) of the categorical predictors and their interaction on the total amounts of esters, alcohols, acids, aldehydes, and ketones produced by *S. cerevisiae* AWRI796 and EC1118 throughout the fermentation of five red grape musts; ns, not significant (*p* > 0.05).

|  | **Must** | **Strain** | **Must × Strain** |
| --- | --- | --- | --- |
| Esters | <0.00001 | <0.00001 | 0.00129 |
| Alcohols | <0.00001 | <0.00001 | 0.00002 |
| Acids | 0.00004 | ns | 0.00217 |
| Aldehydes | 0.00002 | ns | <0.00001 |
| Ketones | <0.00001 | ns | 0.00258 |

More specifically, in Figure 2 the decomposition of the statistical hypothesis for the effects of must (2A), strain (2B) and interaction must × strain (2C) on the total amount of esters is reported. The wines obtained from musts Negroamaro and Magliocco were the samples most affected (2A) from the three variables for ester production and in general *S. cerevisiae* AW was a higher producer of ester compounds (2B, 2C) compared to strain EC in all the samples considered except for must Cs.

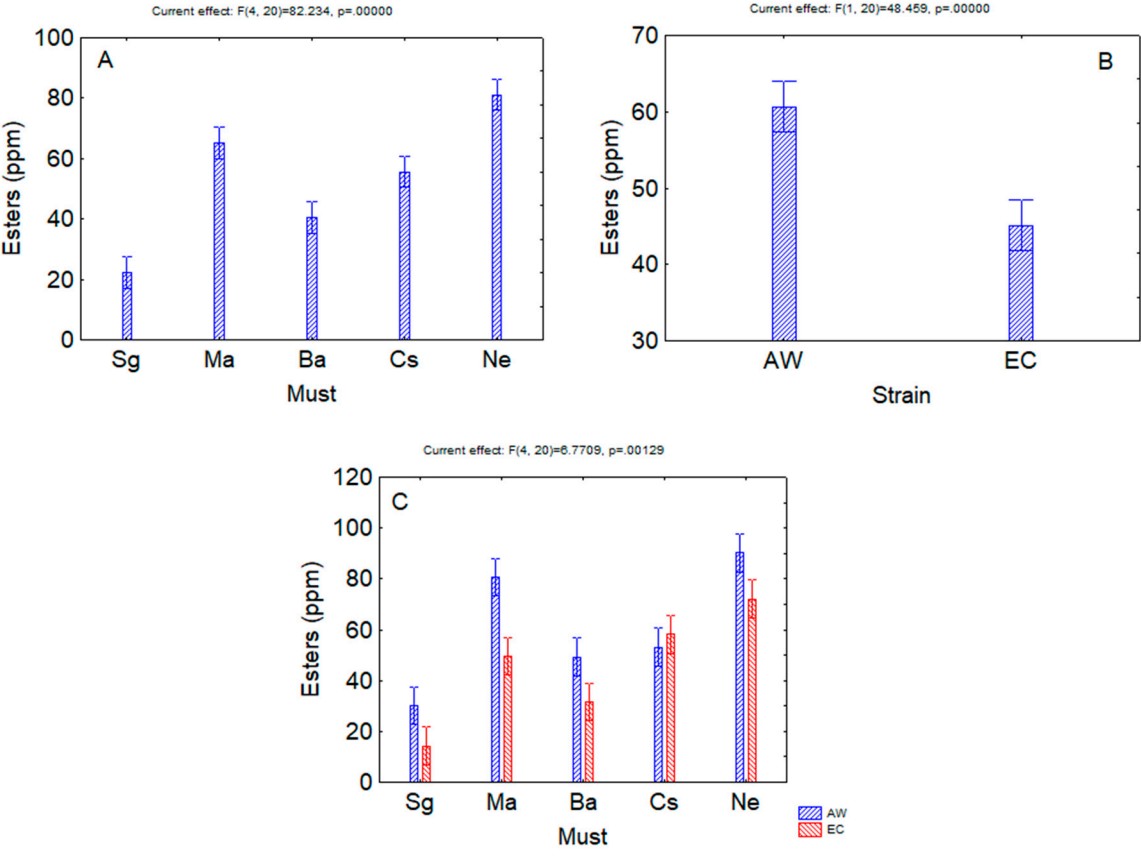

**Figure 2.** Decomposition of the statistical hypothesis for the effects of must (**A**), strain (**B**), and interaction must × strain (**C**) on the total amounts of esters. Bars denote 95%-confidence intervals. Sg, Sangiovese; Ma, Magliocco; Ba, Barbera; Cs, Cabernet-Sauvignon; Ne, Negroamaro. EC, strain EC1118; AW, strain AWRI796.

Figure 3 reports the decomposition of the statistical hypothesis for the effects of must (Figure 3A), strain (Figure 3B) and interaction must × strain (Figure 3C) on the total amounts of alcohols.

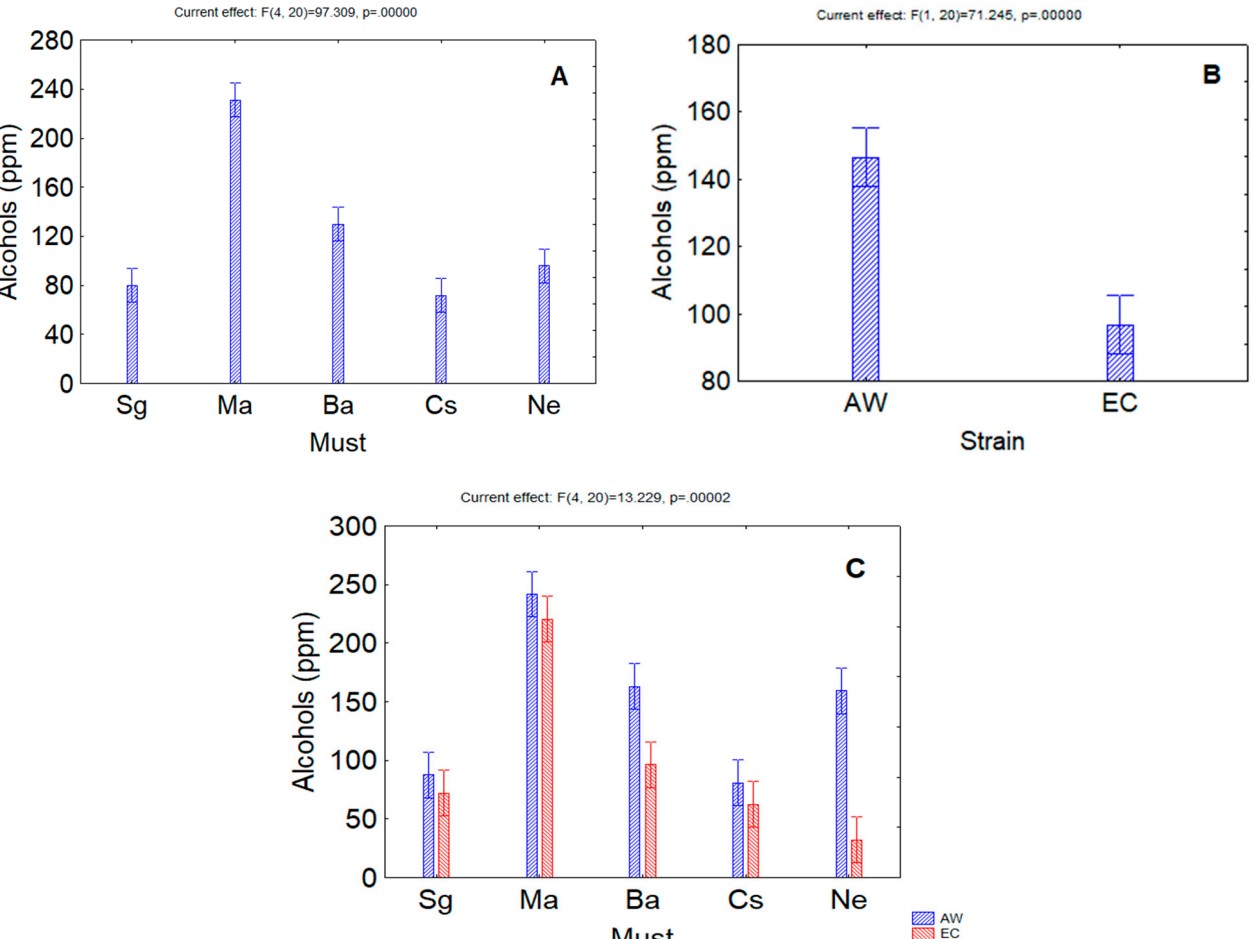

**Figure 3.** Decomposition of the statistical hypothesis for the effects of must (**A**), strain (**B**), and interaction must × strain (**C**) on the total amounts of alcohols. Bars denote 95%-confidence intervals. Sg, Sangiovese; Ma, Magliocco; Ba, Barbera; Cs, Cabernet-Sauvignon; Ne, Negroamaro. EC, strain EC1118; AW, strain AWRI796.

The predictor must affected the samples obtained by the fermentation of Magliocco (Figure 3A), while the influence of the strain *S. cerevisiae* AW on the alcohol production was higher compared to strain EC (Figure 3B). The effect of the interaction was significant in Negroamaro, where the strain AW produced a higher number of alcohols (170 vs. 30 ppm) (Figure 3C).

In Figure 4, the effect of the three variables on acid compounds is reported. Specifically for acid production (Figure 4A), the most affected sample was wine from Sangiovese while the less affected was that one from Barbera, even if the effect of the strain (as variable) was less evident with the exception of sample Negroamaro where the strain *S. cerevisiae* AW produced more acids (Figure 4B).

Since among the volatile organic compounds (VOCs) alcohols and esters represent the most important molecules, some of them were used for further elaborations. In Table 2, the significance of the two variables, must and strain, and their interaction (must × strain) on the amounts of the selected esters and alcohols produced by *S. cerevisiae* AW and EC throughout the fermentation of five red grape musts is reported. The type of must and the strain used in their fermentation, as well as their interaction in the system, were able to significantly affect the amount of the considered esters and alcohols with exception of 3-hydroxy-2,2-dimethoxypropyl acetate and ethyl hexanoate, which were affected by the composition of the initial must and by the interaction must × strain, and hexan-1-ol and 2-methyl-2-nitropropan-1-ol, whose amounts

were affected only by the type of must. On the contrary, propan-1-ol and 2-phenylethanol were not affected by the interaction must × strain used.

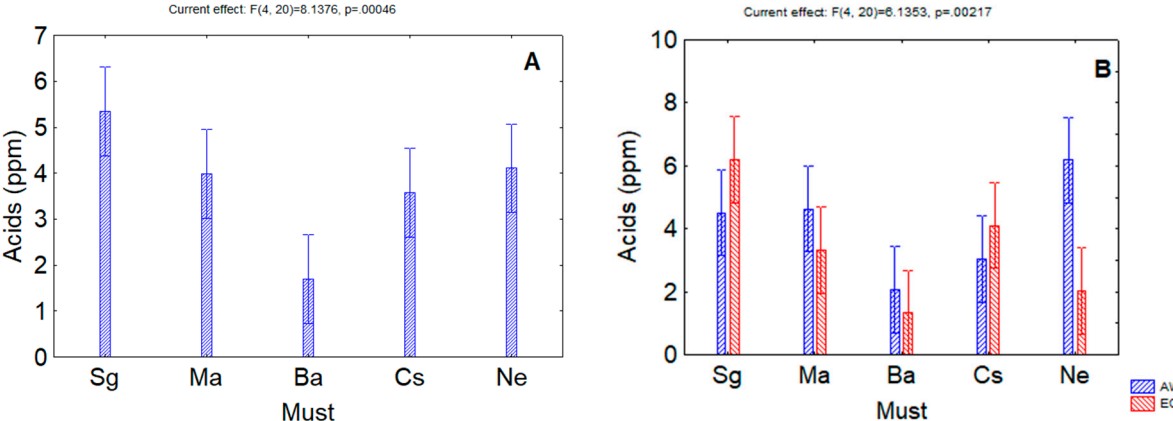

**Figure 4.** Decomposition of the statistical hypothesis for the effects of must (**A**), and interaction must × strain (**B**) on the total amounts of acids. Bars denote 95%-confidence intervals. Sg, Sangiovese; Ma, Magliocco; Ba, Barbera; Cs, Cabernet-Sauvignon; Ne, Negroamaro. EC, strain EC1118; AW, strain AWRI796.

**Table 2.** Significance (*p*-level) of the categorical predictors and their interaction on the amounts of some selected esters and alcohols produced by *S. cerevisiae* AWRI796 and EC1118 throughout the fermentation of five red grape musts; ns, not significant ($p > 0.05$).

| IUPAC Name | Must | Strain | Must × Strain |
|---|---|---|---|
| ethyl undec-10-enoate | <0.00001 | <0.00001 | <0.00001 |
| 3-methylbutyl acetate | <0.00001 | 0.00993 | 0.00396 |
| [(2E)-3,7-dimethylocta-2,6-dienyl] acetate | <0.00001 | 0.00182 | 0.00002 |
| methyl non-7-ynoate | 0.00004 | 0.00251 | 0.00004 |
| 2-phenylethyl acetate | <0.00001 | 0.00113 | <0.00001 |
| 3-hydroxy-2,2-dimethoxypropyl acetate | <0.00001 | ns | 0.03931 |
| ethyl decanoate | <0.00001 | 0.00818 | 0.00094 |
| ethyl dodecanoate | <0.00001 | 0.00036 | <0.00001 |
| ethyl acetate | <0.00001 | <0.00001 | 0.00006 |
| ethyl hexanoate | 0.00001 | ns | 0.00053 |
| ethyl octanoate | <0.00001 | <0.00001 | <0.00001 |
| 2-methylbutane-1,3-diol | <0.00001 | 0.00008 | <0.00001 |
| 3-methylbutan-1-ol | <0.00001 | <0.00001 | <0.00001 |
| hexan-1-ol | 0.00066 | ns | ns |
| propan-1-ol | 0.000485 | 0.00004 | ns |
| 2-methylpropan-1-ol | <0.00001 | 0.00991 | 0.00192 |
| 2-methyl-2-nitropropan-1-ol | <0.00001 | ns | ns |
| 3-methylsulfanylpropan-1-ol | <0.00001 | 0.00004 | <0.00001 |
| 3-methylhexan-3-ol | <0.00001 | <0.00001 | <0.00001 |
| 4-(methoxymethoxy)-3-nitropentan-2-ol | <0.00001 | <0.00001 | <0.00001 |
| 5-methylsulfanyl-3H-1,3,4-thiadiazole-2-thione | <0.00001 | <0.00001 | <0.00001 |
| [2-(2-aminopropoxy)-3-methylphenyl] methanol | <0.00001 | 0.00009 | <0.00001 |
| 2-phenylethanol | <0.00001 | 0.00043 | ns |

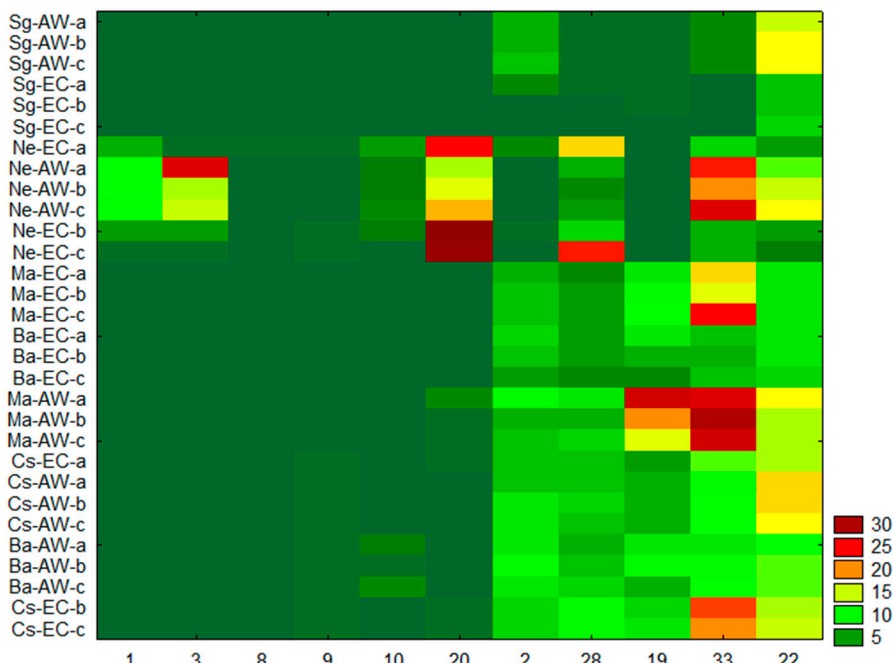

**Figure 5.** Two way-joining run on some selected esters (ppm). Sg, Sangiovese; Ma, Magliocco; Ba, Barbera; Cs, Cabernet-Sauvignon; Ne, Negroamaro. EC, strain EC1118; AW, strain AWRI796. 1: ethyl undec-10-enoate; 2: 3-methylbutyl acetate; 3: [(2E)-3,7-dimethylocta-2,6-dienyl] acetate; 8: methyl non-7-ynoate; 9: 2-phenylethyl acetate; 10: 3-hydroxy-2,2-dimethoxypropyl acetate; 19: ethyl decanoate; 20: ethyl dodecanoate; 22: ethyl acetate; 28: ethyl hexanoate; 33: ethyl octanoate.

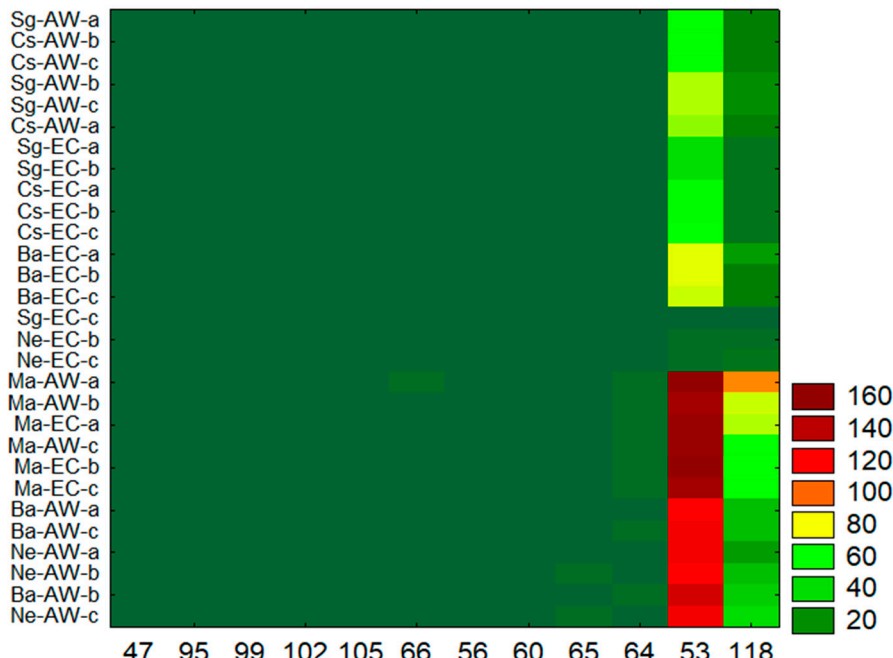

**Figure 6.** Two way-joining run on some selected alcohols (ppm). Sg, Sangiovese; Ma, Magliocco; Ba, Barbera; Cs, Cabernet-Sauvignon; Ne, Negroamaro. EC, strain EC1118; AW, strain AWRI796. 47: 2-methylbutane-1,3-diol; 53: 3-methylbutan-1-ol; 56: hexan-1-ol; 60: propan-1-ol; 64: 2-methylpropan-1-ol; 65: 2-methyl-2-nitropropan-1-ol; 66: 3-methylsulfanylpropan-1-ol; 95: 3-methylhexan-3-ol; 99: 4-(methoxymethoxy)-3-nitropentan-2-ol; 102: 5-methylsulfanyl-3H-1,3,4-thiadiazole-2-thione; 105: [2-(2-aminopropoxy)-3-methylphenyl] methanol; 118: 2-phenylethanol.

Due to the large dataset of information acquired, the raw volatile compound data, used in the previous elaboration, were analyzed by means of PCA to pinpoint the effects of the strains AW and EC. In Figure 7, the projection of the detected classes of compounds (Figure 7A) and the samples (Figure 7B), as mean of the three independent repetitions, is reported. As shown in PCA, which explains more than 65% of variance among the samples considering all the classes of compounds, the effect of the must, the strain, and their interaction is evident. For instance, the type of strain applied was able to affect the separation of Magliocco wines on the factor plan along the PC2 while this was less evident for Cabernet-Sauvignon wines.

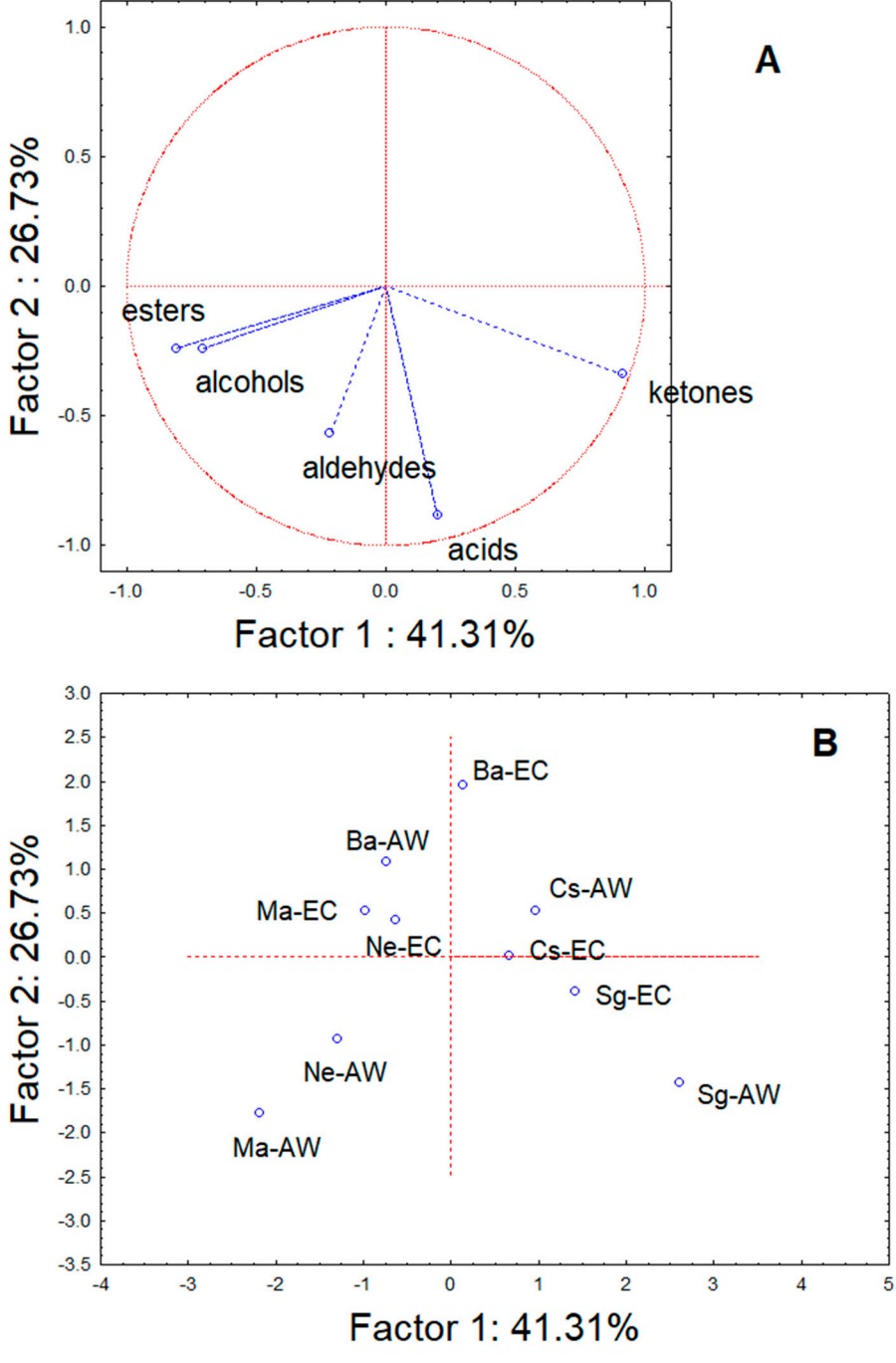

**Figure 7.** Principal component analysis run on the total amounts of esters, alcohols, acids, aldehydes, and ketones. (**A**) Variable projection; (**B**) case projection. Sg, Sangiovese; Ma, Magliocco; Ba, Barbera; Cs, Cabernet-Sauvignon; Ne, Negroamaro. EC, strain EC1118; AW, strain AWRI796.

Considering esters and alcohols separately, Figure 8 reports the projection of esters (Figure 8A) and wine samples (Figure 8B) on the factor planes PC1 and PC2 that describe the 42.57 and 24.89% of variance among the samples, respectively. The effect of the strains is less evident except for wine obtained from Negroamaro. This sample was clearly separated from all the others, meaning that the initial must has an influence on ester production. Similar results were obtained for alcohols (Figure 9), since wines obtained from must Negroamaro were well separated on the factor plan (Figure 9A). Alcohols that allowed this cluster formation were: 3-methylhexan-3-ol, hexan-1-ol and 4-(methoxymethoxy)-3-nitropentan-2-ol (Figure 9B).

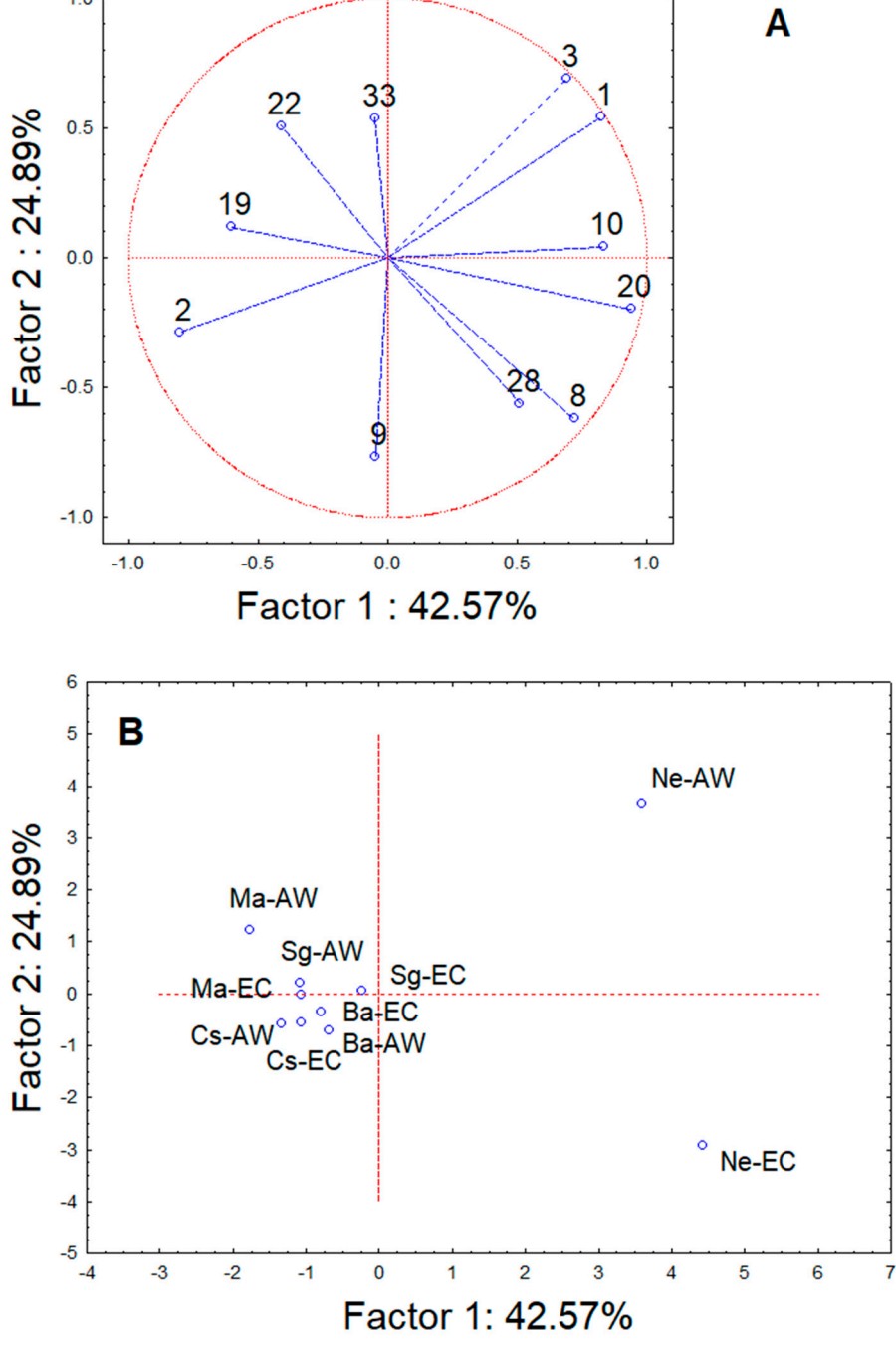

**Figure 8.** Principal component analysis run on some selected esters. (**A**) Variable projection; (**B**) case projection. Sg, Sangiovese; Ma, Magliocco; Ba, Barbera; Cs, Cabernet-Sauvignon; Ne, Negroamaro. EC,

strain EC1118; AW, strain AWRI796. 1: ethyl undec-10-enoate; 2: 3-methylbutyl acetate; 3: [(2E)-3,7-dimethylocta-2,6-dienyl] acetate; 8: methyl non-7-ynoate; 9: 2-phenylethyl acetate; 10: (3-hydroxy-2,2-dimethoxypropyl) acetate; 19: ethyl decanoate; 20: ethyl dodecanoate; 22: ethyl acetate; 28: ethyl hexanoate; 33: ethyl octanoate.

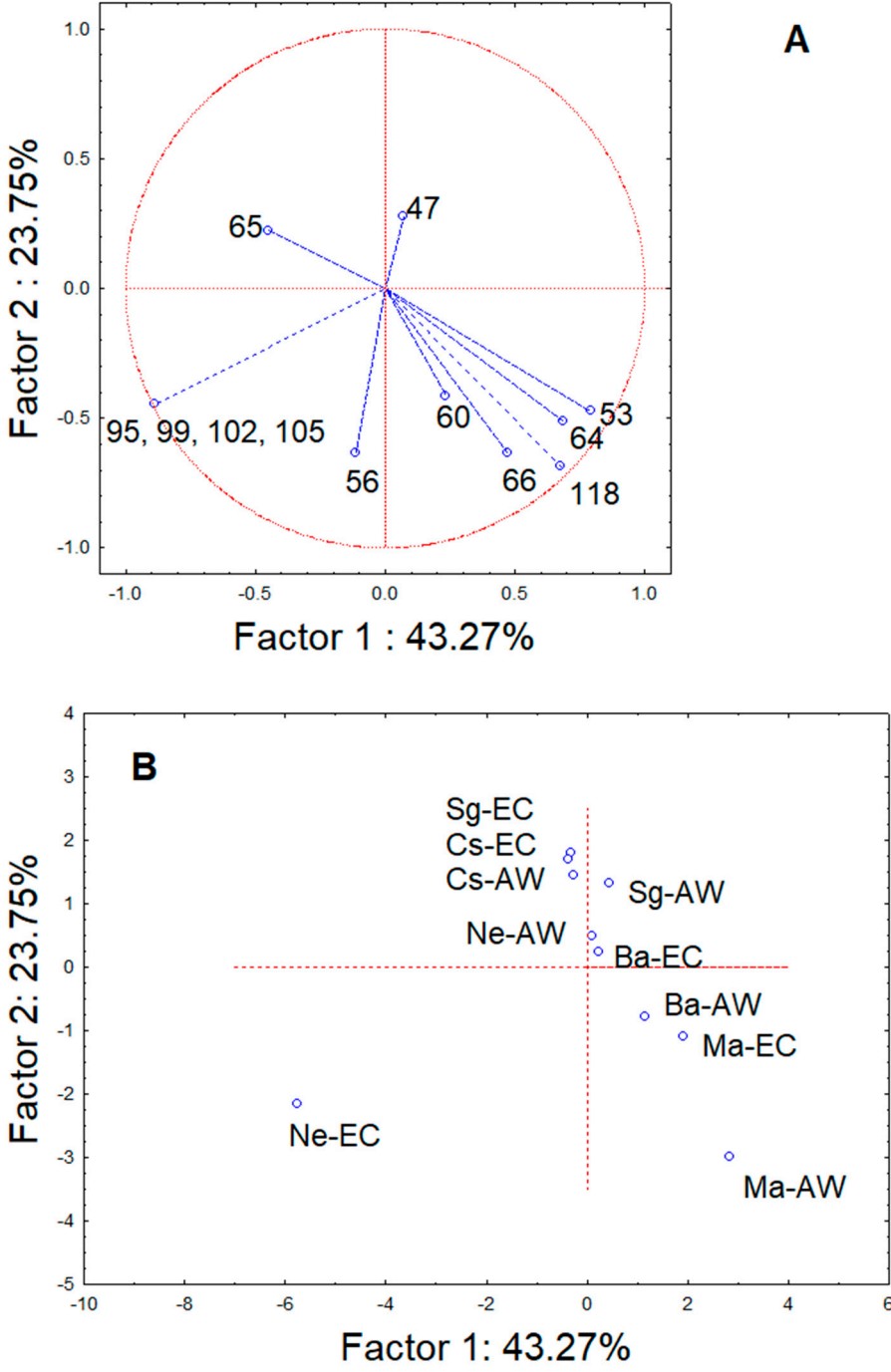

**Figure 9.** Principal component analysis run on some selected alcohols. (**A**) Variable projection; (**B**) case projection. Sg, Sangiovese; Ma, Magliocco; Ba, Barbera; Cs, Cabernet-Sauvignon; Ne, Negroamaro. EC, strain EC1118; AW, strain AWRI796. 47: 2-methylbutane-1,3-diol; 53: 3-methylbutan-1-ol; 56: hexan-1-ol; 60: propan-1-ol; 64: 2-methylpropan-1-ol; 65: 2-methyl-2-nitropropan-1-ol; 66: 3-methylsulfanylpropan-1-ol; 95: 3-methylhexan-3-ol; 99: 4-(methoxymethoxy)-3-nitropentan-2-ol; 102: 5-methylsulfanyl-3H-1,3,4-thiadiazole-2-thione; 105: [2-(2-aminopropoxy)-3-methylphenyl] methanol; 118: 2-phenylethanol.

## 4. Discussion

Wines produced by each RU were analyzed at the end of fermentation by means of SPME-GC/MS to detect VOCs. Several studies reported the benefit of using the SPME-GC/MS technique to provide volatile molecule fingerprinting of food and beverages in relation to their microbiota and/or production processes [38,42,43].

Considering the main aromatic components, the identified VOC profiles of the five wines were mainly affected by grape must, indicating a greater varietal influence. This confirms that the metabolism of inoculated strains can be significantly impacted by small changes in must composition, altering, as a result, the formation of fermentative aroma compounds, as already reported by other authors [44,45]. In fact, grape varietals have a significant impact on the formation of yeast metabolites because they possess distinct amino acid profiles that can be used as precursors for the formation of fusel alcohols and esters [46]. At the same time, other compounds (such as polyphenols) can induce a lower cell growth, with higher mortality in the early stage of the process, and an increased fermentation time [47]. The effects of non-aroma compounds of the matrix, such as precursors of terpenes and thiols, have been also shown to be important determinant factors for the final aroma profile of wine [48]. Finally, grape variety still has a significant impact on naturally occurring grape microbiota persistence during the fermentation process that can differently interact with *Saccharomyces* strains used as starters [49].

In our study, alongside a strong influence of grape must, strain and strain/must interaction also played a significant role in the final wine aroma profile. *S. cerevisiae* AW exerted a greater effect mainly on esters and alcohols, compared to the use of strain EC. This was true particularly for Magliocco and Negroamaro musts fermented with AW where the highest concentration of esters and alcohols was observed, respectively. Similar results were reported by Casu et al. [50] who described a higher concentration of acetate esters and alcohols in Sauvignon blanc wines made with AW instead of using strain EC. Other authors reported the impact of two commercial strains on the level of thiol 3-mercaptohexanol (3MH) and its acetylated derivative 3-mercaptohexyl acetate (3MHA) in response to must composition. For example, a higher amount of linoleic acid caused a 100% reduction in 3MHA concentration when grape must was fermented by *S. cerevisiae* EC compared to its corresponding control, while a 69% reduction in 3MHA levels was observed in wines fermented by *S. cerevisiae* AW [51]. The same authors found that must supplemented with linoleic acid promoted a 17% increase in 3MH concentration when using the strain EC, while the same substrate was promoting a reduction in 3MH by 14% with the other strain. In another study regarding Arinto white wines [52], EC produced wines characterized by esters, such as ethyl decanoate, decyl acetate, and ethyl laurate, and alcohols, such as butyl alcohol, were compared to wines produced with other strains. Since wines were fermented under the same conditions and the initial must was the same, this result demonstrates that yeasts have different metabolisms and therefore the final wines have different characteristic aroma profiles, as reported for other wine varieties [53–57]. In our study, the effect of the applied strains is less evident on the final wine composition, with differences mainly related to the grape variety. Our changes were more quantitative than qualitative within each type of wine. Similarly, in Riesling and Vidal ice wines, strain EC affected the odor-active compounds, but the effects of the yeast depended mainly on the cultivar and the vintage [58]. Negroamaro wines (sample Ne) are an exception. In fact, in this case, PCA analysis clustered the musts fermented with EC and AW in two separate quadrants, showing that the yeast strain played an important and fundamental role on the definition of Negroamaro aromatic component.

Looking at single components, medium chain fatty acid ethyl esters were particularly produced in a strain dependent fashion. For instance, ethyl decanoate and ethyl octanoate were detected at a higher amount in fermented musts of Magliocco or Negroamaro using the strain AW, while ethyl dodecanoate was largely produced in the Negroamaro must fermented with both the strains. Even Ricker et al. [46] reported higher amounts of ethyl decanoate and ethyl hexanoate in Chardonnay wines obtained with AW compared to those

made with *S. cerevisiae* EC. These compounds are responsible for the highly desired fruity, candy, and perfume-like aromas of fermented beverages, and due to their lower threshold values, compared to other aroma compounds, can strongly impact sensory quality of wine. They can even synergize each other, acting well below their threshold values [59–61].

## 5. Conclusions

The results obtained in the framework of the joint experiments promoted by the Italian Group of Microbiology of Vine and Wine (GMVV) could have several impacts for winemaking research. From a practical point of view, they confirm the importance of using a suitable and robust protocol, and standardized conditions to study the performances of starter cultures, thus avoiding the confounding effect of methodological variables (inoculation, tools to assess fermentation, conditions of the assay, etc.). Moreover, they demonstrate that the five final wines analyzed for VOCs were mainly affected by the grape used and the interaction with the *Saccharomyces* strain applied. For instance, some musts could produce wines with higher levels of aroma compounds, e.g., esters in Negroamaro, and alcohols in Magliocco, while strain AW could contribute to increasing the level of these compounds, with respect to strain EC. Nevertheless, the most important outcome of this research is in regards to the importance of the interaction strain x must, mainly on single compounds (ethyl decanoate, ethyl octanoate, and others), suggesting that the description of some strains at a general level (high-production of VOCs, esters, alcohols) could be coupled with the description at molecule level, with some compounds more affected than others and useful as targets or indicators to define strain imprinting.

**Author Contributions:** Conceptualization, P.R., A.B., F.P. and R.L.; methodology, P.R. and A.B.; software, A.B. and B.S.; validation, P.R., A.B. and F.P; formal analysis, A.B.; investigation, D.G., G.S., I.V., A.T., V.E., G.B., F.G. and B.S.; data curation, A.B., F.P., D.G. and G.S.; writing—original draft preparation, P.R., A.B. and F.P.; writing—review and editing, F.P, G.S., D.G., I.V., A.T., V.E., G.B., F.G., R.L., B.S., A.B. and P.R.; visualization, F.P.; supervision, P.R., A.B. and F.P. All authors have read and agreed to the published version of the manuscript.

**Funding:** This research received no external funding.

**Data Availability Statement:** The data presented in this study are available in the article.

**Conflicts of Interest:** The authors declare no conflict of interest.

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
