# Peer review of "Impact of Two Commercial S. cerevisiae Strains on the Aroma Profiles of Different Regional Musts"

_beverages, doi:10.3390/beverages8040059_

Round 1

Reviewer 1 Report

While the manuscript is relatively well written, the experiment is not novel. The experimental design is also fundamentally flawed. Without sterilization the grape musts will contain yeasts from the vineyard which are known to influence fermentations, particularly in the early stages. The authors do not address this at all. Their conclusions regarding the interaction of yeast x variety are therefore potentially inaccurate. It is unfortunate that this was not considered as the rest of the experiment was carefully planned and executed.

Author Response

While the manuscript is relatively well written, the experiment is not novel. The experimental design is also fundamentally flawed. Without sterilization the grape musts will contain yeasts from the vineyard which are known to influence fermentations, particularly in the early stages. The authors do not address this at all. Their conclusions regarding the interaction of yeast x variety are therefore potentially inaccurate. It is unfortunate that this was not considered as the rest of the experiment was carefully planned and executed.

We thank the reviewer for his/her consideration. The main objective of our work was to couple a certain aromatic result with a behaviour typical of the inoculated S. cerevisiae strain in different red grape musts. Focusing on the aroma, we could not sterilize the grape musts because the sterilization is not a common practice in winemaking as it influences the must composition with the loss of some precursors, present in the grape must, which play a fundamental role in the formation of the compounds responsible for the aromatic profile of wine. For this reason, SO2 and massive inoculum of S. cerevisiae strains (2 x 106 cell/mL) were added to avoid the development of spontaneous microbiota. After strain inoculation, the presence of viable cells was verified by plate counting on WL medium, and already at the beginning of the fermentation the results confirmed the total dominance of the inoculated S. cerevisiae strain.

Our experimental design was planned to eliminate all the process variables in order to obtain reliable results on the interaction of S. cerevisiae strain and grape variety. In comparison to all the data available in literature, the novelty of our work is the assessment of the differential impact of two commercial S. cerevisiae wine strains on the aroma profiling of five different regional grape musts by inter-laboratory scale comparative fermentations carried out by five different research units (RUs) under identical conditions and following the same protocol during the entire process. Hoping to have clarified our experimental design, we improved these points throughout the manuscript.

We have improved these points adding all these information in the text.

Reviewer 2 Report

My biggest criticism of the paper is simple: I would like to see the authors promote the work done in the paper a little more clearly to help entice the reader.

Suggested Revisions:

Grammar:

Remove the words ‘Background’, ‘Methods’, and ‘Conclusions’ from the abstract (these would be considered headings, and while instructions for authors specify to use those or the organization, it does state not to include them).

L102 'being the main responsible' appears to be missing a word

L112 'related on' should be 'related to'

Please check capitalization of figure is consistent (L280 and L371 uncapitalized). 

L417 'variable' should be 'variables'

General comment: please consider removing/decreasing the unnecessary qualifying words at the start of sentences such as therefore, in fact, moreover, etc. In L64-67 specifically, therefore is used twice in a row, it makes it difficult to read. Presenting the information without these words is more than doable; please remove any immediate repeats (as in L64-L67 as mentioned above, another example in L492-495 with the repetition of ‘moreover’) and consider decreasing on use of these words in the piece as a whole. 

Small comments: 

Please introduce Figure 9 in the text. 9A and 9B are referenced but Figure 9 itself is never introduced. Suggested rewording of sentence in 376 to accomplish this goal: 'Similar results were obtained for alcohols, demonstrated in Figure 9, for the wines obtained…'

Please review the sentence beginning on L495, 'Moreover, the effect of strains acted in...' as the final part of the sentence, ‘the strain effect in general way is a no-sense in wine microbiology’ is difficult to understand and needs to be rewritten for clarity. 

Remaining comments:

In the introduction, please elaborate more on what makes this study unique. The methodology is from a previous study; what makes this study interesting? Why should people read this work? Please elaborate more on what is unique about this work, why it is important, and what the emphasis on the study is in the final section of the introduction. 

For Figure 3, please include the must x strain plot present in Figures 2 and 4. Based on the results in Figure 4B, I can understand not including a strain plot for the acids, but from Figures 3A and 3B, I can’t understand the choice not to include the must x strain plot in this case.

For the conclusion, please consider elaborating more on the results from the paper. You mention that ‘the variable strain could have a different weight, with some musts experiencing a different trend depending on the strain’ but while the details are in the paper, this is an example of an interesting trend that could be explored further in separate works. It reads as though the emphasis in this paper is the protocol of the analysis and less the results themselves, but the protocol is already published elsewhere based on the phrasing in the introduction. Highlight the interesting aspects of this paper. Emphasize what findings make this paper unique.  

Author Response

My biggest criticism of the paper is simple: I would like to see the authors promote the work done in the paper a little more clearly to help entice the reader.

We have followed the reviewer’s suggestion, adding more specific information and improving the introduction regarding the purpose of our work.

Suggested Revisions:

Grammar:

Remove the words ‘Background’, ‘Methods’, and ‘Conclusions’ from the abstract (these would be considered headings, and while instructions for authors specify to use those or the organization, it does state not to include them).

We have modified the abstract.

L102 'being the main responsible' appears to be missing a word

We have added “compound”

L112 'related on' should be 'related to'

We have modified as suggested.

Please check capitalization of figure is consistent (L280 and L371 uncapitalized). 

We have checked the capitalization of the figures.

L417 'variable' should be 'variables'

We have deleted the whole sentence in the text.

General comment: please consider removing/decreasing the unnecessary qualifying words at the start of sentences such as therefore, in fact, moreover, etc. In L64-67 specifically, therefore is used twice in a row, it makes it difficult to read. Presenting the information without these words is more than doable; please remove any immediate repeats (as in L64-L67 as mentioned above, another example in L492-495 with the repetition of ‘moreover’) and consider decreasing on use of these words in the piece as a whole. 

 We have removed the unnecessary qualifying words throughout the manuscript.

Small comments: 

Please introduce Figure 9 in the text. 9A and 9B are referenced but Figure 9 itself is never introduced. Suggested rewording of sentence in 376 to accomplish this goal: 'Similar results were obtained for alcohols, demonstrated in Figure 9, for the wines obtained…'

The Figure 9 has been introduced in the text.

Please review the sentence beginning on L495, 'Moreover, the effect of strains acted in...' as the final part of the sentence, ‘the strain effect in general way is a no-sense in wine microbiology’ is difficult to understand and needs to be rewritten for clarity. 

We have eliminated this sentence and rewritten the concept.

Remaining comments:

In the introduction, please elaborate more on what makes this study unique. The methodology is from a previous study; what makes this study interesting? Why should people read this work? Please elaborate more on what is unique about this work, why it is important, and what the emphasis on the study is in the final section of the introduction.

We have modified the introduction, adding more specific information regarding the purpose and novelty of our work.

For Figure 3, please include the must x strain plot present in Figures 2 and 4. Based on the results in Figure 4B, I can understand not including a strain plot for the acids, but from Figures 3A and 3B, I can’t understand the choice not to include the must x strain plot in this case.

We have added the must x strain plot (Figure 3C).

For the conclusion, please consider elaborating more on the results from the paper. You mention that ‘the variable strain could have a different weight, with some musts experiencing a different trend depending on the strain’ but while the details are in the paper, this is an example of an interesting trend that could be explored further in separate works. It reads as though the emphasis in this paper is the protocol of the analysis and less the results themselves, but the protocol is already published elsewhere based on the phrasing in the introduction. Highlight the interesting aspects of this paper. Emphasize what findings make this paper unique.  

The data reported in the literature regarding the aroma composition of wines produced by different S. cerevisiae strains refer to tests performed in a single laboratory and therefore it is difficult to compare the data, due to the differences in protocols and process variables applied in the various laboratories. The uniqueness of our work is the evaluation of the differential impact of two commercial wine strains of S. cerevisiae on the aromatic profile of five different regional grape musts by inter-laboratory scale comparative fermentations carried out by five different research units (RUs) under identical conditions and following the same protocol throughout the entire process. Furthermore, to eliminate the variable linked to the inoculation of the strain, the same batch of each commercial yeast was used for all the fermentations carried out by the five RUs. To the best of our knowledge, similar experiments have not been performed and are not available in literature. We do hope to have clarified better the purpose and the novelty of our work. We have implemented the text.

Reviewer 3 Report

The topic of the manuscript is interesting and a lot of work was done but there are some things that have to be corrected in order to improve manuscript quality.

1. The title of the article is very long and to some extend unclear

2. Introduction is too long but incomplete. There is no need to be so detailed about S. cerevisiae (ln. 48-60). According to me it will be better to write something about  the other metabolites that are produced by yeast (aldehydes, organic acid, ketones) because you comment them in Results and Discussion

3. Materials and methods - it will be good to divide 2.1. Sample production in subsections. For example: 2.1.1 Microorganisms 2.1.2. Yeast preparation, and etc. It will be easier for reading

Please, explain why you have chosen 25C for fermentation temperature.

Please, explain how you have determine sugar depletion and YAN.

Results - Table 2 is very unclear. It will be better to write the name of compounds in the Table, not to use their numbers

Write the IUPAC names of the compounds 10, 28, 56, and etc.

Discussion - A lot of things written in Discussion are not for this section but for Introduction. The substantial part of Discussion is from ln 442 till the end.

Author Response

The topic of the manuscript is interesting and a lot of work was done but there are some things that have to be corrected in order to improve manuscript quality.

  1. The title of the article is very long and to some extend unclear

We have changed the title in “Impact of two commercial S. cerevisiae strains on the aroma profiles of different regional musts”.

  1. Introduction is too long but incomplete. There is no need to be so detailed about S. cerevisiae (ln. 48-60). According to me it will be better to write something about the other metabolites that are produced by yeast (aldehydes, organic acid, ketones) because you comment them in Results and Discussion

We have modified the introduction, eliminating the first part, inserting some sentences from the Discussion and information about organic acid, aldehydes and ketones, as suggested by the reviewer

  1. Materials and methods - it will be good to divide 2.1. Sample production in subsections. For example: 2.1.1 Microorganisms 2.1.2. Yeast preparation, and etc. It will be easier for reading

We have modified the section Materials and Methods, subdividing in subsections, as suggested by the reviewer

Please, explain why you have chosen 25C for fermentation temperature.

Considering the influence of the process variables, including temperature, on the expression of aromas by the yeast, we decided to set the temperature to 25 °C as it can be considered the optimal temperature for S. cerevisiae wine strains.

Please, explain how you have determine sugar depletion and YAN.

In the section Materials and Methods we have added the determination of sugar depletion and YAN

Results - Table 2 is very unclear. It will be better to write the name of compounds in the Table, not to use their numbers

Thank you for the comment, Table 2 was adjusted accordingly

Write the IUPAC names of the compounds 10, 28, 56, and etc.

Thank you for the comment, the names were introduced using the IUPAC nomenclature

Discussion - A lot of things written in Discussion are not for this section but for Introduction. The substantial part of Discussion is from ln 442 till the end.

We have eliminated from the discussion the part indicated and have implemented the discussion.

Round 2

Reviewer 1 Report

The edits have sufficiently addressed concerns regarding the experimental design. Some further revision for grammar would be useful.

Author Response

We thank the reviewer for his/her suggestion and we carefully checked the grammar throughout the text.

Reviewer 3 Report

The authors have taken into account all the reviewers' comment, which has led to the improvement of the manuscript quality.

Author Response

The authors are thankful to the reviewer for his/her valuable suggestions, which were very useful for improving our paper.